

# Comprehensive transcriptome analysis of peripheral blood unravels key lncRNAs implicated in ABPA and asthma

Chen Huang[1,2], Dongliang Leng[3], Peiyan Zheng[4], Min Deng[3], Lu Li[4], Ge Wu[4], Baoqing Sun[4] and Xiaohua Douglas Zhang[3]

[1] Dr. Neher's Biophysics Laboratory for Innovative Drug Discovery, Macau University of Science and Technology, Macau, SAR, China, Macau, China
[2] Stat Key laboratory of Quality Research in Chinese Medicine, Macau Institute For Applied Research in Medicine and Health, Macau University of Science and Technology, Macau, SAR, China, Macau, China
[3] Faculty of Health and Science, University of Macau, Macao, Macau
[4] Department of Allergy and Clinical Immunology, State Key Laboratory of Respiratory Disease, National Clinical Research Center of Respiratory Disease, Guangzhou Institute of Respiratory, Health, First Affiliated Hospital of Guangzhou Medical University, Guangzhou, Guangdong, China

Corresponding author
Xiaohua Douglas Zhang,
douglaszhang@um.edu.mo

## ABSTRACT

Allergic bronchopulmonary aspergillosis (ABPA) is a complex hypersensitivity lung disease caused by a fungus known as *Aspergillus fumigatus*. It complicates and aggravates asthma. Despite their potential associations, the underlying mechanisms of asthma developing into ABPA remain obscure. Here we performed an integrative transcriptome analysis based on three types of human peripheral blood, which derived from ABPA patients, asthmatic patients and health controls, aiming to identify crucial lncRNAs implicated in ABPA and asthma. Initially, a high-confidence dataset of lncRNAs was identified using a stringent filtering pipeline. A comparative mutational analysis revealed no significant difference among these samples. Differential expression analysis disclosed several immune-related mRNAs and lncRNAs differentially expressed in ABPA and asthma. For each disease, three sub-networks were established using differential network analysis. Many key lncRNAs implicated in ABPA and asthma were identified, respectively, i.e., AL139423.1-201, AC106028.4-201, HNRNPUL1-210, PUF60-218 and SREBF1-208. Our analysis indicated that these lncRNAs exhibits in the loss-of-function networks, and the expression of which were repressed in the occurrences of both diseases, implying their important roles in the immune-related processes in response to the occurrence of both diseases. Above all, our analysis proposed a new point of view to explore the relationship between ABPA and asthma, which might provide new clues to unveil the pathogenic mechanisms for both diseases.

## INTRODUCTION

Allergic bronchopulmonary aspergillosis (ABPA) is a Th2 hypersensitivity disease response to the presence of *Aspergillus fumigatus* in the bronchial mucosa. The first case was described by *Hinson, Moon & Plummer (1952)* in the United Kingdom (*Hinson, Moon & Plummer, 1952*). The prevalence of ABPA was initially thought to be rare, which was estimated to be occurred in 1~2% in asthmatic patients (*Singh et al., 2018*). However, increasing evidences indicate that ABPA is underdiagnosed and much more prevalent than previously estimated (*Greenberger, 2002*; *Patel et al., 2019*). *Denning, Pleuvry & Cole (2013)* estimates that global prevalence of ABPA may be 0.7–3.5% of asthmatic patients, but now be increased into 2.5% (*Patel et al., 2019*). On the other hand, asthma is a common lung condition characterized by bronchial hyper-responsiveness and variable airflow obstruction. A few cases exhibit complicated and uncontrolled symptoms. This type of asthma is defined as severe asthma, which could be closely linked to atopic to airborne allergens, especially to fungal spores (*Agarwal, 2011*). As one of main risk factor for ABPA, *Aspergillus fumigatus* is the most common fungi causing fungal sensitization (*Agarwal, 2011*; *Agarwal & Gupta, 2011*), suggesting close association between ABPA and asthma.

In spite of the fact that the correlation between asthma and ABPA has been well elucidated, the detailed causality between two entities, particularly the pathogenesis of ABPA is not fully understood yet. For instance, only a small proportion of severe asthmatic patients develop ABPA. Both diseases are characterized by extremely high serum level of IgE (Immunoglobulin E) but a small proportion of ABPA patients has less high serum IgE level (*Agarwal et al., 2019*). Approximately 60% of ABPA patients can benefit from traditional treatment for allergic fungal disorder, e.g., the use of systemic oral itraconazole, but a few patients exhibit uncontrolled symptoms despite various treatments have been tried.

Recently, some progress has been made in disclosing genetic polymorphisms associated with ABPA risk and progression. A typical example indicates that asthmatic patients expressing HLA-DR2 and/or DR5 and lacking HLA-DQ2 are susceptible to develop ABPA after exposure to *A. fumigatus* (*Chauhan et al., 2000*). Around half of non-ABPA atopic *Aspergillus*-sensitive individuals are found to possess HLA-DR2 and/or HLADR5 genotype, whereas the HLA-DRB1*1501 and HLA-DRB1*1503 genotype are reported to have high relative risk (*Knutsen, 2017*). The mutations in cystic fibrosis transmembrane conductance regulator gene (CFTR) is reported to raise the risk of ABPA in asthmatic patients (*Agarwal et al., 2012*).

Despites there are some advances in the disclosure of genetic risk underlying ABPA, there are still gaps to be filled. Typically, ABPA patients are found to harbor high frequency of polymorphism in Toll-like receptor 9T-1237C (TLR9T-1237C). However, the polymorphisms of TLR9 are reported to have no associations with the patients with severe asthma who are also associated with Aspergillus sensitivity (*Carvalho et al., 2008*). It was suggested that genetic variations are not the only factor contributing ABPA pathogenesis. Increasing evidences demonstrated that transcriptome dysfunction and
aberrant gene expression also have a critical role in disease pathogenesis (*Zeller et al., 2010*). In particular, lncRNAs are proved to be key regulatory layers associated with asthma. For instance, lncRNAs BCYRN1 is found to promote the proliferation and migration of rat airway smooth muscle cells in asthma via activation of transient receptor potential one (*Zhang et al., 2016*). Here, aiming to identify potential lncRNAs implicated in the occurrence of ABPA and asthma, as well as their possible roles in both diseases, we performed a comprehensive transcriptome analysis on the three different types of human peripheral blood, including ABPA, asthma and health controls. Our analysis discovered many key lncRNAs which are likely to be related to immune functions and might play key roles in the pathogenesis of ABPA and asthma. Our findings benefit the discovery of novel biomarkers and targets guiding for diagnosis and therapy for both diseases.

## MATERIALS AND METHODS

### Inclusion, sampling and deep RNA sequencing

This study was approved by the Medical Ethics Committee of First Affiliated Hospital of Guangzhou Medical University (ethics approval no. gyfyy-2016-73). All experiments were performed in accordance with relevant guidelines and regulations of the Ethics Committee of First Affiliated Hospital of Guangzhou Medical University. All participants provided written informed consent prior to the publication of clinical and sequencing data. Briefly, a total of 27 unique male individuals were enrolled for our study, including 7 asthma patients (diagnosed as allergic asthma), 12 ABPA patients and 8 healthy individuals considered as control group (Table 1). The diagnosis of asthma was according to the latest Global Initiative for Asthma (GINA) guidelines, and the diagnosis of ABPA was based on the criteria of The International Society of Human and Animal Mycology (ISHAM) working group, which contains two obligatory criteria and three additional criteria (*Shah & Panjabi, 2016*). Based on the clinical symptoms, 5 asthma patients were at exacerbation stage and two were at chronic stage (Table S1).

And according to the proposed clinical staging of ABPA (*Shah & Panjabi, 2016*), among the included ABPA patients, seven were at acute stage, one was at exacerbation stage, two was at response stage, while the other two were not able to assess the clinical stage because of insufficient clinical data (Table S1).

For each enrolled subject, the peripheral whole blood was extracted and the peripheral blood mononuclear cell (PBMC) was separated immediately by ficoll-paque. The total RNA was extracted using trizol (invitrogen) method and the library for RNA sequencing was prepared based on a standard protocol established by RiboBio Company in Guangzhou and sequenced on Illumina. All the raw RNA-seq data used in the present study are deposited at Short Read Archive (SRA) database of NCBI (https://trace.ncbi.nlm.nih.gov/Traces/sra/) and are assigned the accession number PRJNA582337.

### Raw data processing

The raw data was trimmed by Trimmomatic v0.36 (ILLUMINACLIP: TruSeq3-PE.fa:2:30:10:8:true SLIDINGWINDOW:4:15 LEADING:3 TRAILING:3 MINLEN:50)
**Table 1  Baseline characteristics of the study population.**

|  | Asthma (*n* = 7) | ABPA (*n* = 12) | Healthy control (*n* = 8) | *P* value |
|---|---|---|---|---|
| Age (years) | 44 [41, 46] | 34 [29, 40.75]* | 26.50 [23.25, 35.75]* | 0.007 |
| Blood cell count |  |  |  |  |
| Eosinophil (×10⁹ cells/L) | 0.36 [0.05, 0.98] | 0.58 [0.33, 0.90] | 0.20 [0.10, 0.35] | 0.284 |
| Neutrophil (×10⁹ cells/L) | 5.30 [4.00, 7.50] | 5.00 [3.40, 7.40] | 3.60 [3.20, 4.35] | 0.208 |
| Induced sputum |  |  |  |  |
| Neutrophil (%) | 79.00 [47.00, 88.50] | 39.75 [14.25, 76.88] | – | 0.297 |
| Macrophage (%) | 9.00 [7.80, 12.00] | 33.75 [0.13, 79.75] | – | 0.999 |
| Eosinophil (%) | 11.50 [0.50, 40.50] | 8.00 [1.38, 39.00] | – | 0.999 |
| Lymphocytes (%) | 1.00 [0.50, 1.80] | 1.50 [0.50, 5.13] | – | 0.655 |
| Immunological characteristic |  |  |  |  |
| Total IgE level (kUA/L) | 529 [40.2, 826] | 2595 [695.5, 4757]* | – | 0.040 |
| sIgE positivity (%) | 57.1% | 100%* | – | 0.013 |
| *A.f* sIgE positivity (%) | 14.3% | 100%** | – | <0.001 |
| Glucocorticoids |  |  |  |  |
| Use of oral glucocorticoids (%) | 42.9% | 58.3% | – | 0.515 |
| Use of inhaled glucocorticoids (%) | 71.4% | 50% | – | 0.361 |

**Note:**
Data were given as medians with interquartile range (IQR).
* *P* < 0.05 compared with asthma.
** *P* < 0.01 compared with asthma.

(*Bolger, Lohse & Usadel, 2014*) and the cleaned reads were aligned against Ensembl hg38 human genome via STAR (v020201) (*Dobin et al., 2013*). The transcriptome was re-constructed using StringTie (v1.3.3b) (*Pertea et al., 2015*). A stringent stepwise pipeline (Fig. 1) which had been applied in our previous studies (*Huang et al., 2019a*; *Leng et al., 2019*; *Zheng et al., 2020*) was utilized for identification of high-confidence dataset of lncRNAs.

## Identification of lncRNAs

Firstly, the known lncRNAs were picked out from complete set of assembled transcripts according to the "biotype_transcript" of reference gtf file of *Homo sapiens* from Ensembl database (*Zerbino et al., 2018*). The transcript was detected as lncRNA when its "biotype_transcript" tagged as "Long_non-coding_RNA", "Non_coding", "3prime_overlapping_ncRNA", "Antisense", "lincRNA", "Retained_intron", "Sense_intronic", "Sense_overlapping", "macro_lncRNA" and "bidirectional_promoter_lncRNA".

Then the human known mRNAs derived from Ensembl database were excluded from the remaining transcripts. Afterwards, the alignment was performed for the remaining transcripts against known protein sequences from NCBI nr database (*Pruitt, Tatusova & Maglott, 2007*) and Uniprot database (*UniProt Consortium, 2018*). Successfully aligned transcripts were excluded from potentially non-protein-coding sequences. For the unmapped transcripts, filtering was conducted to get rid of the sequences with length less than 200 nt and the longest ORF longer than 100 residues. Finally, the qualified sequences
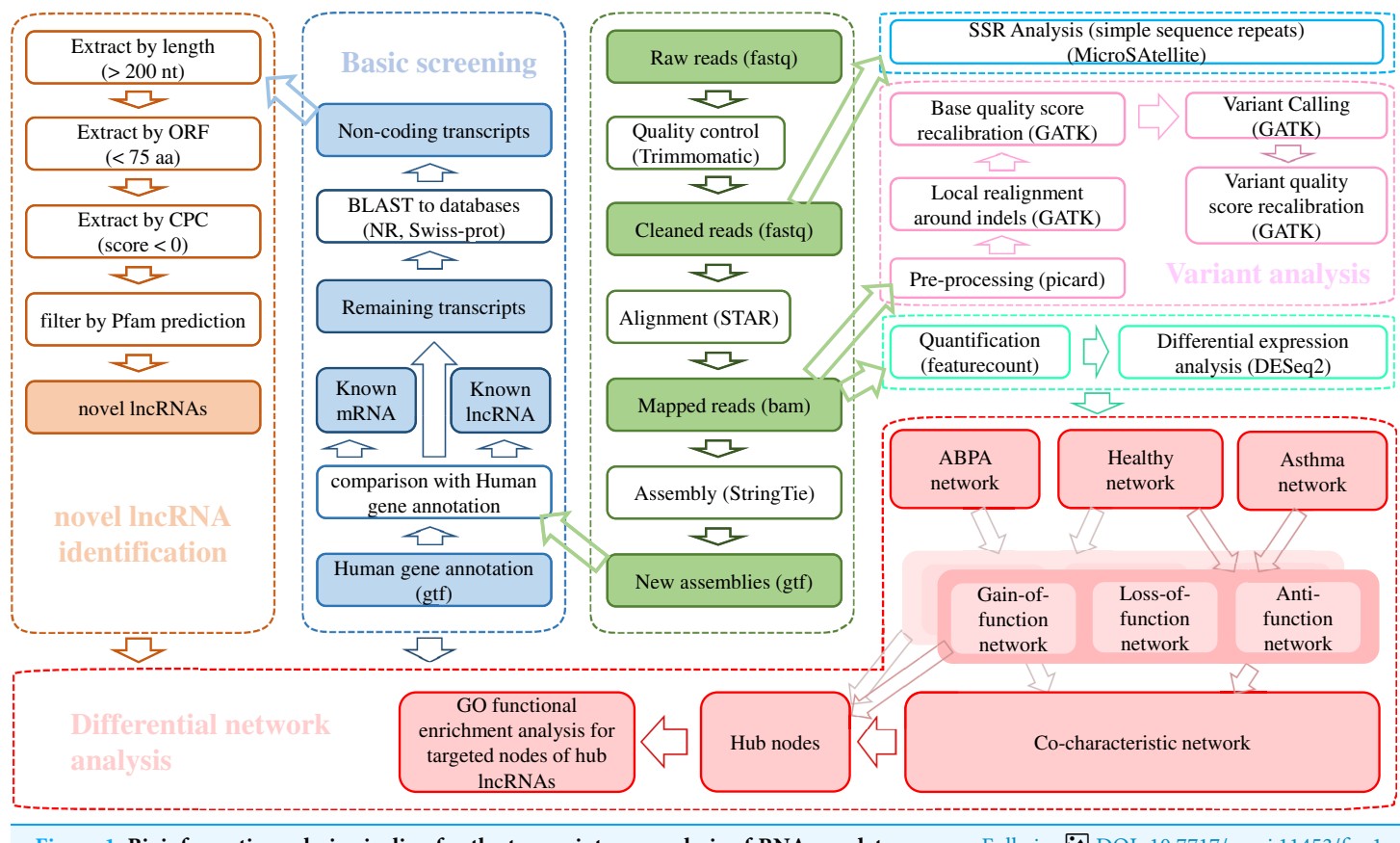

**Figure 1** Bioinformatic analysis pipeline for the transcriptome analysis of RNA-seq data.

were further filtered to remove the protein-coding sequences according to the results from Pfamscan (*El-Gebali et al., 2019*) and CPC (*Kong et al., 2007*). Eventually, the remaining transcripts were reserved as final dataset of lncRNAs.

## Transcriptome variations analysis

For single nucleotide variants, i.e., SNPs and small insertion and deletion (INDEL) detection, we applied Genome Analysis Toolkit (GATK) analysis pipeline (*Van der Auwera et al. 2013*) for variant calling. All the variants were annotated by ANNOVAR (*Wang, Li & Hakonarson, 2010*). Simple sequence repeats (SSRs) detection was performed using the Perl script of MISA–MicroSAtellite identification tool (http://pgrc. ipk-gatersleben.de/misa/) with default parameters. Statistical hypotheses were used to investigate whether there were significant differences in the observed proportion of SNPs, INDELs and SSRs among populations of ABPA, asthma and healthy control. The detailed process sees Supplementary Methods.

## Transcriptome expression analysis

Differential expression analysis was performed using R package DESeq2 (adjusted *p*-value ≤ 0.05) (*Love, Huber & Anders, 2014*). Functional enrichment analysis was performed via ClueGO V2.5.2 (*Bindea et al., 2009*) plugin in Cytoscape V3.6.1 (*Shannon et al., 2003*),

which could generate a dynamical network structure composed of functionally grouped terms based on the gene list of interest, i.e., differentially expressed genes. The differentially expressed transcripts were subjected to R programming language for hierarchical cluster analysis to detect distinct expression patterns.

Differential interaction network analysis was conducted based on three filtering criteria, which generated three sub-networks for each group, including loss-of-function network, gain-of-function network, and anti-function network. A total of 12 algorithms (MCC, DMNC, MNC, Degree, EPC, BottleNeck, EcCentricity, Closeness, Radiality, Betweenness, Stress, ClusteringCoefficient) of cytoHubba (Chin et al., 2014) plugin in Cytoscape were used to identify hub node for the established network. Detailed procedure see Supplementary Methods. For the hub node validation, the expression profile of four data sets (GSE35571, GSE473, GSE31773, GSE2125) from NCBI GEO database were downloaded and analyzed (Table S2).

## Statistics of clinical data

For the basic statistics of clinical data (File S3), all the data were shown in median (interquartile range). Non-parametric Kruskal–Wallis test was performed to compare the differences between three groups while Mann–Whitney test was performed to compare the differences between two groups. Regarding the ratio data, Chi-square test was applied for the comparison. Statistical analysis was performed by SPSS 22.0. A value of $P < 0.05$ was considered as statistically significant.

# RESULTS

## Characteristics of study subjects

As shown in Table 1, the asthma group was elder compared with either ABPA group or Healthy control group ($P < 0.05$). There was no significant difference in eosinophil or neutrophil counts of peripheral blood in the three groups. Regarding the percentages of inflammatory cells in induced sputum, no statistical difference was found in any type of inflammatory cells between asthma and ABPA. There was also no statistical difference in the results of lung function between the two patient groups. In the comparison of IgE levels, patients with ABPA had higher total IgE levels ($P < 0.05$), higher positive rate of specific IgE ($P < 0.05$), in particular *A. fumigatus* specific IgE ($P < 0.01$) than patients with asthma. In addition, use of glucocorticoids (either oral or inhaled) did not differ significantly between asthma and ABPA.

## LncRNAs identification

RNA-seq via Illumina sequencing platform yielded on average 130 million reads per sample (Table S3). After quality trimming, the clean reads were mapped to human genome and subsequently assembled into 283,883 transcripts. A total of 94,155 lncRNAs across 27 samples, including 52,791 known lncRNAs and 41,364 novel lncRNAs were detected using a stringent stepwise filter pipeline (Fig. 1).

**Table 2 Basic statistics of variants occurred in RNA-seq data detected by GATK analysis pipeline.**

| Variant type | Number |
|---|---|
| UTR5; UTR3 | 55 |
| UTR5 | 17,091 |
| UTR3 | 147,976 |
| upstream; downstream | 2,484 |
| upstream | 39,078 |
| downstream | 63,520 |
| splicing | 2,523 |
| exonic | 62,680 |
| exonic; splicing | 53 |
| intergenic | 1,359,937 |
| intronic | 3,170,032 |
| ncRNA_exonic | 33,883 |
| ncRNA_exonic; splicing | 22 |
| ncRNA_intronic | 293,956 |
| ncRNA_splicing | 177 |
| ncRNA_UTR5 | 1 |

## No significant difference in variants across ABPA, asthma and health controls

A total of 4,521,799 SNPs and 671,669 INDELs were identified across 27 samples via GATK analysis pipeline (Fig. 1, Table S4). Majority of variants locates in intronic (~61%) and intergenic regions (~26.2%) (Table 2, Fig. S1). A total of 62,733 variants are in genic regions, and 42.9% of them are synonymous SNP (Table S5). 44,090 variants significantly existed differentially across groups by Fisher Exact probability test (File S1). In addition, many genes (including few lncRNAs) exhibited distinct variants cross groups (Fig. 2A).

For SSRs analysis, a total of 4,740,915 different types of SSRs were identified (175,589 per sample on average) (Table S6). No significant difference in SSRs across these samples was detected (Figs. 2D–2I), despites two statistical approaches (details process see Supplementary Methods) revealed around 150 SSRs were differentially enriched in total (Figs. 2B, 2C, File S2).

## Differential expression analysis reveals immune-related processes implicated in both diseases

A total of 2,481 differentially expressed transcripts (DETs, including 554 lncRNAs) were found in the comparison of ABPA vs healthy controls (Figs. 3A–3D), and 2,706 transcripts (including 529 lncRNAs) were differentially expressed in the comparison of asthma vs healthy controls, respectively. Functional analysis via ClueGO for both DET datasets demonstrated that many immune-related biological processes were significantly enriched, including leukocyte activation, activation of innate immune response and positive

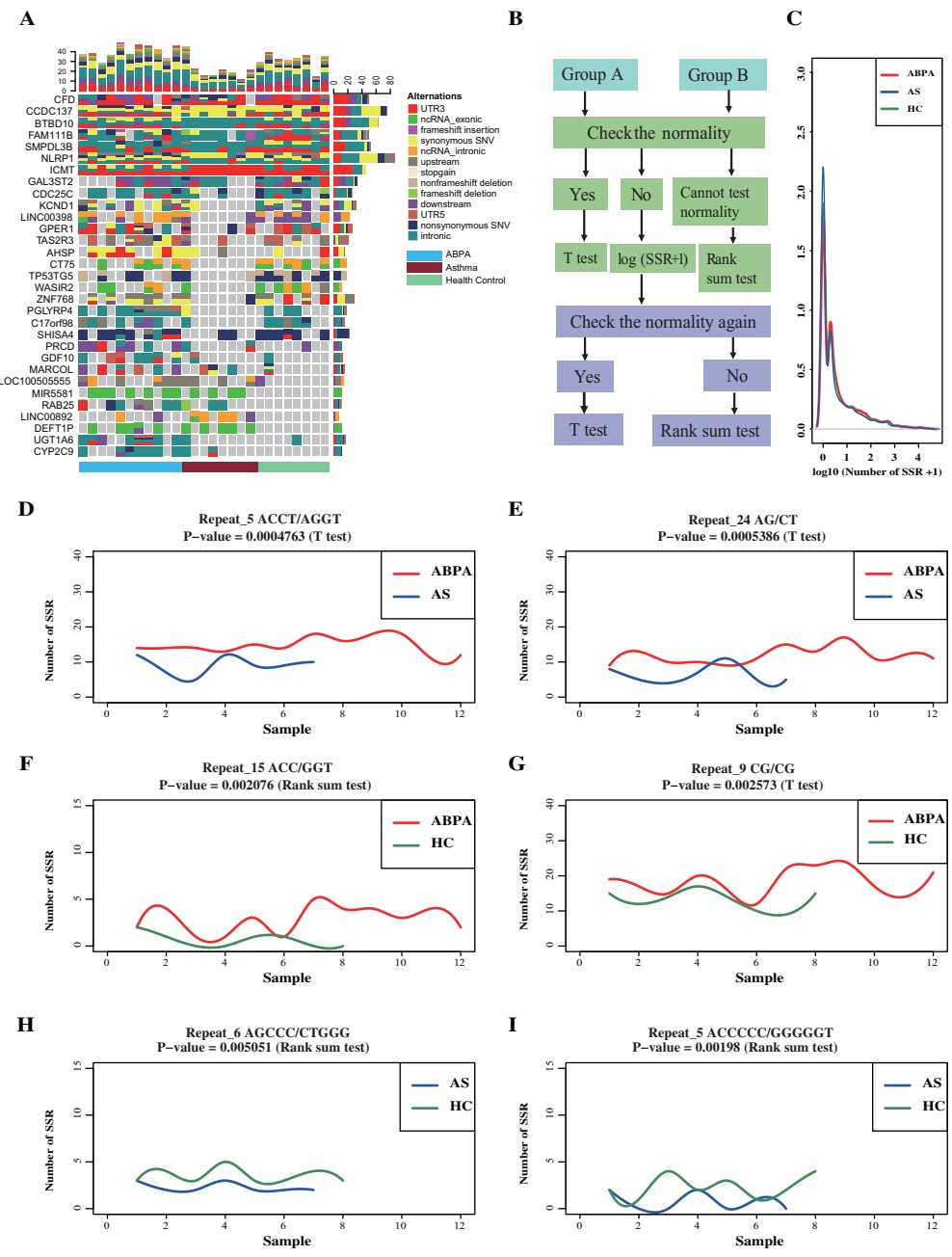

**Figure 2 Transcriptome-level mutational landscape of ABPA and asthma compared to health controls.** (A) Each row in the figure corresponds to one gene/lncRNA, whereas each column corresponds to one sample ($n = 28$). (blue bar represents ABPA patients, dark red bar represents asthma patients and light green represents health controls). (Top) Bar plots describing the percentage of different type of alternations identified in each sample across all the identified genes/lncRNAs. (Right) Bar plots of the percentage of different type of alternations of each gene/lncRNAs across all the sample. (Bottom) bar plot represents the type of samples, blue bar represents ABPA patients, dark red bar represents asthma patients and light green represents health controls. (B) The statistical analysis pipeline to detect whether the identified SSR event significantly existed between ABPA patients and health controls, as well as asthma patients and health controls. (C) The density distribution of the SSRs among the three types of samples. The graphs (D~I) plot the SSRs difference between ABPA patients, asthma patients as well as health controls.

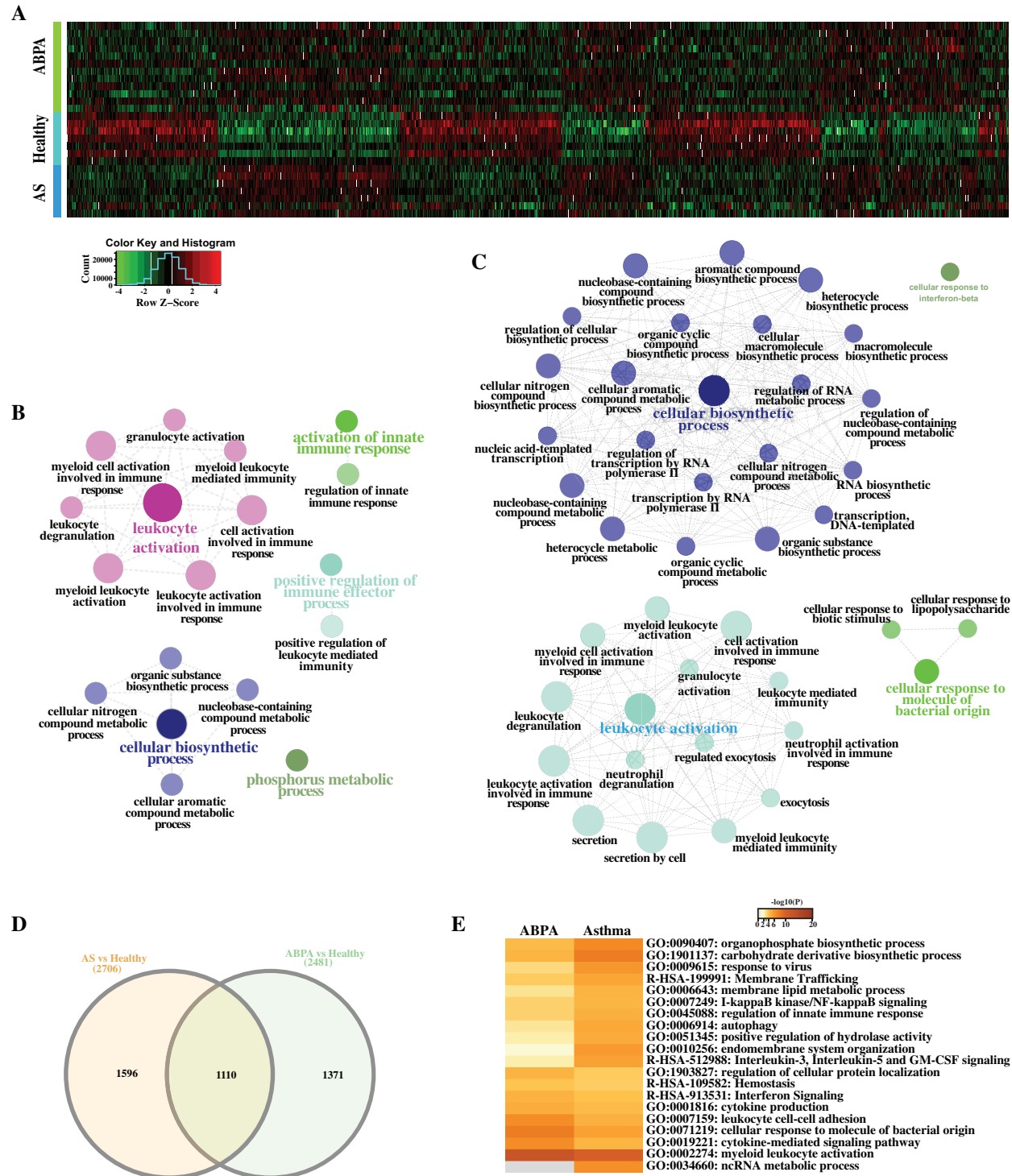

**Figure 3 Differential expression analysis of ABPA, asthma patients and health controls.** (A) Heatmap of all the differentially expressed transcripts identified by the comparison of ABPA with health controls and asthma with health controls. (B) Functional enrichment analysis based on the differentially expressed transcripts identified by the comparison of asthma with health controls. (C) Functional enrichment analysis based on the differentially expressed transcripts identified by the comparison of ABPA with health controls. (D) Overlap of the differentially expressed transcripts between two comparisons. (E) Functional enrichment analysis of the differentially expressed transcripts for both diseases.

regulation of immune effector process, etc. (Figs. 3B, 3C). Furthermore by the comparison of two DET datasets, distinct DETs involved in diverse biological processes were detected among three different groups (Figs. 3D, 3E).

Additionally, hierarchical cluster analysis via R language based on DETs revealed three main expression patterns of DETs cross these samples (Fig. 4). Concretely, many transcripts (including mRNAs and lncRNAs) are found to be down-regulated in healthy controls but were up-regulated in ABPA and asthma patients (Fig. 4A). Functional analysis showed that they might be involved in myeloid leukocyte mediated immunity, regulation of defense response to virus, etc. (Fig. 4B). On the contrary, several transcripts are down-regulated in healthy controls but are up-regulated in ABPA and asthma patients (Fig. 4E), which might be associated with columnar/cuboidal epithelial cell differentiation, regulation of apoptotic signaling pathway, etc. (Fig. 4F). A few transcripts are specifically up-regulated in ABPA patients (Fig. 4C), and their functions might be correlated to supramolecular fiber organization and positive regulation of organelle organization (Fig. 4D).

## Differential network analysis reveals crucial lncRNAs correlated to both diseases

The aforementioned analysis has revealed several lncRNAs and mRNAs differentially expressed in ABPA and asthma patients. Next to investigate the possible roles of these lncRNAs implicated in the pathogenesis of ABPA and asthma, we proposed a differential network analysis. Initially, we built an RNA–RNA interaction network based on the expression profile data of eight healthy controls using the Pearson Correlation Coefficient (PCC) method. Then the interactions of these RNAs in the networks were re-assessed based on the expression profiles of 12 ABPA patients and 7 asthmatic patients. To assess the changes of network from healthy status to disease status, three different filtering criteria (details see Supplementary Methods) was proposed to construct three sub-networks for each disease, namely loss-of-function network, gain-of-function network, and anti-function network.

In loss-of-function network, for instance, 293,926 RNA–RNA interactions with strong correlation ($|PCC value| \geq 0.70$ & $P$-value $< 0.05$) are found in health controls, whereas these interactions are shown having weak correlation ($|PCC value| \leq 0.30$) in ABPA patients. It is suggested these RNA–RNA interactions might be disrupted by the occurrence of ABPA. 243,399 RNA–RNA interactions exhibit similar situation in asthma. For each disease, the immune-related RNA–RNA interactions in two loss-of-function networks were extracted to establish the loss-of-immune-related-function network, respectively (Figs. 5A, 5B). Key elements (hub nodes of the network) in both networks were detected by topological network analysis via Cytohubba. Our analysis revealed several hub lncRNAs which might play key roles in the immune dysfunction of ABPA and asthma. In this way, we built the gain-of-function and anti-function networks for both diseases (Figs. 5C–5F), and relevant key lncRNAs and mRNAs were disclosed (Table S7, S8).

To further investigate convincing RNA–RNA interactions implicated in ABPA and asthma, we integrated the above-mention networks (Fig. S2). Notably, the RNAs in the

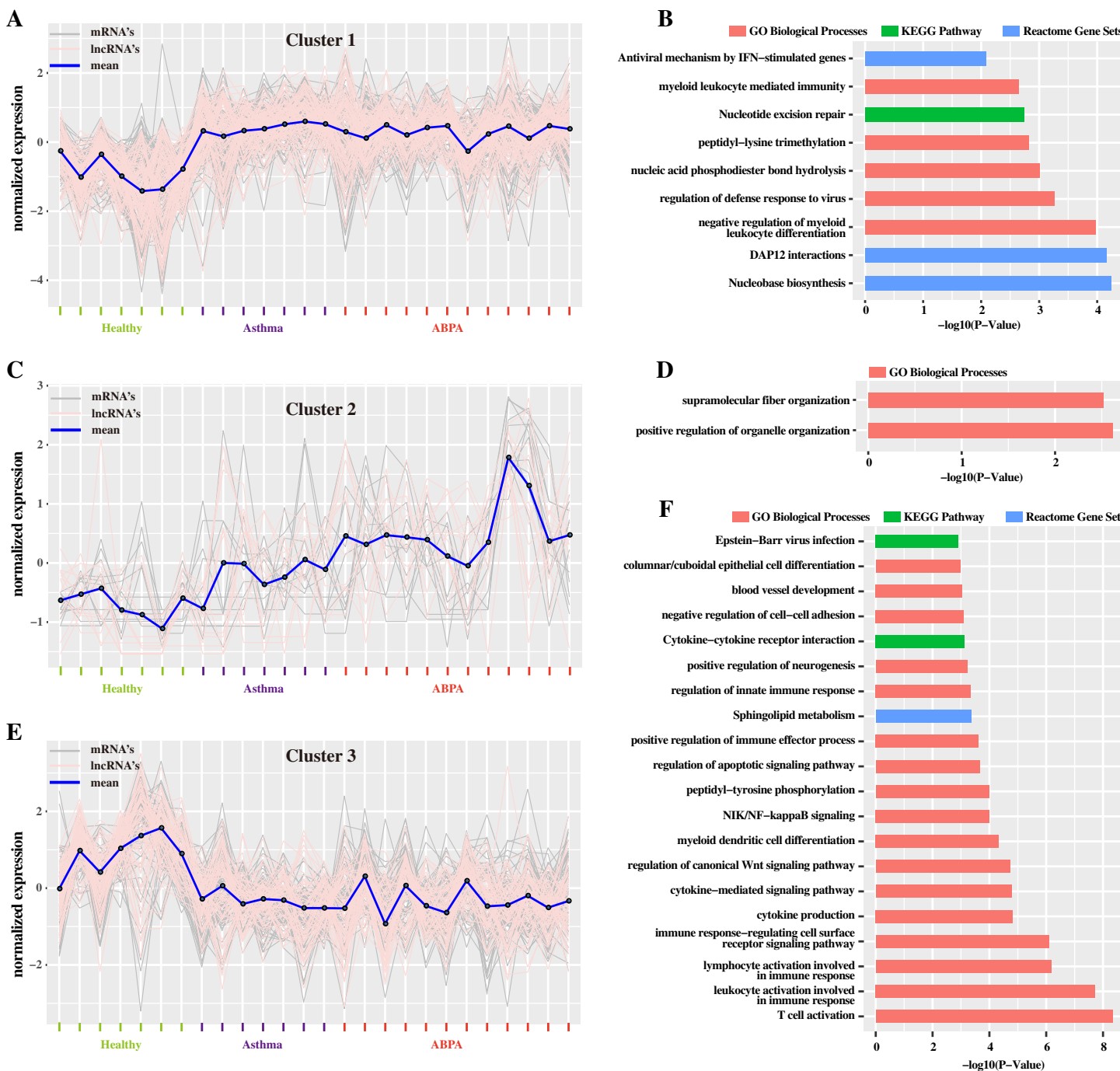

**Figure 4 Cluster analysis of differentially expressed mRNAs and lncRNAs in all human blood samples.** (A) Cluster 1 indicates the mRNAs and lncRNAs that were down-regulation expressed in health controls but were up-regulation in ABPA and asthma patients. (B) Bar plot shows the Gene ontology (GO) functional enrichment analysis based on the mRNAs of cluster 1. (C) Cluster 2 indicates the mRNAs and lncRNAs that specifically up-regulation expressed in ABPA patients. (D) Bar plot shows the Gene ontology (GO) functional enrichment analysis based on the mRNAs of cluster 2. (E) Cluster 3 indicates the mRNAs and lncRNAs that were up-regulation expressed in health controls but were down-regulation in ABPA and asthma patients. (F) Bar plot shows the Gene ontology (GO) functional enrichment analysis based on the mRNAs of cluster 3.

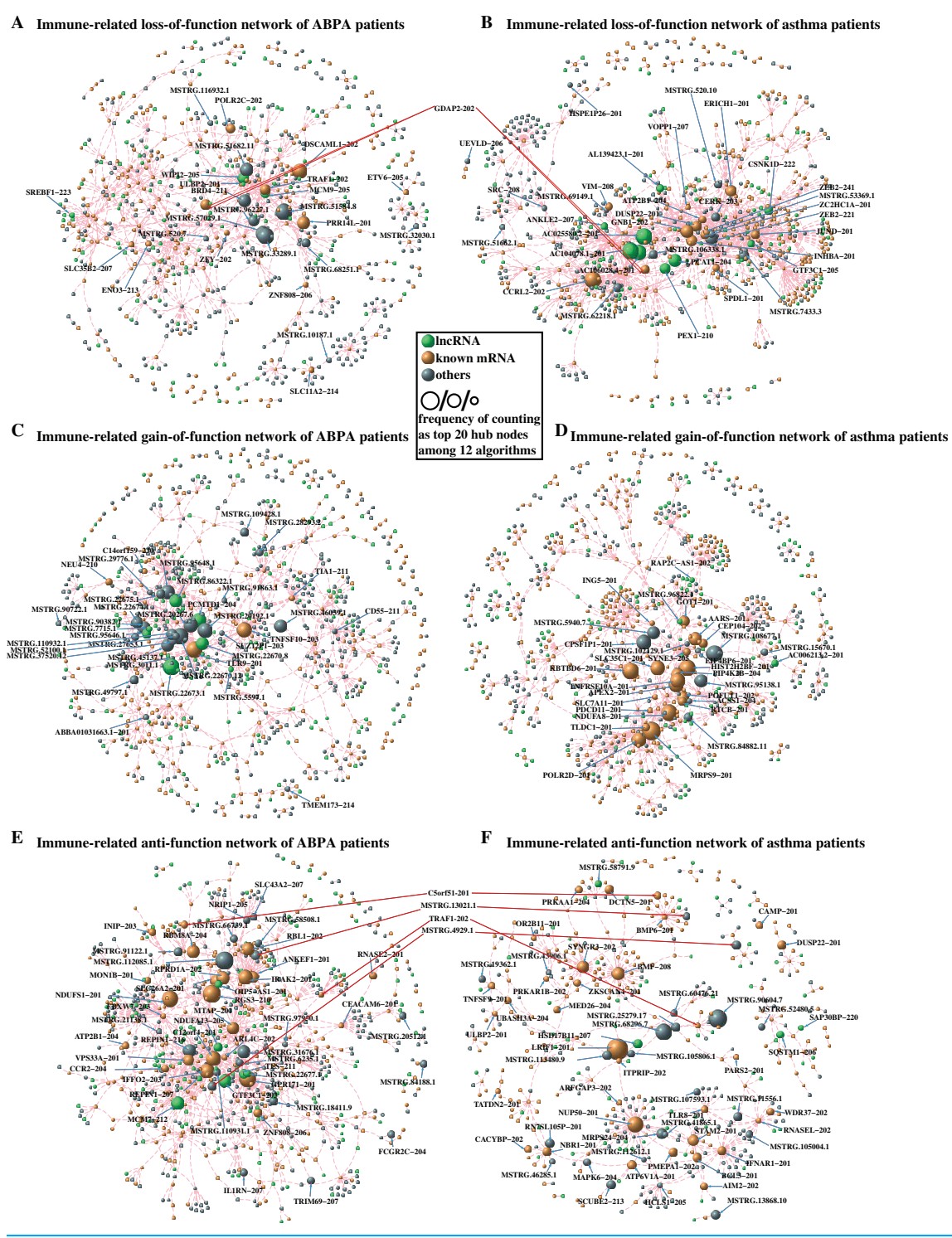

**Figure 5 Differential network analysis between ABPA, asthma patients and health controls.** (A) Visualization of the immune-related loss-of-function network of ABPA patients ($|PCC_{healthy}| \geq 0.95$, $|PCC_{ABPA}| \leq 0.30$, $|PCC_{ABPA - healthy}| \geq 1.00$). (B) Visualization of the immune-related loss-of-function network of asthma patients ($|PCC_{healthy}| \geq 0.93$, $|PCC_{AS}| \leq 0.30$, $|PCC_{AS - healthy}| \geq 1.00$). (C) Visualization of the immune-related gain-of-function network of ABPA patients ($|PCC_{healthy}| \leq 0.30$, $|PCC_{ABPA}| \geq 0.80$, $|PCC_{ABPA - healthy}| \geq 1.00$). (D) Visualization of the immune-related gain-of-function network of asthma patients ($|PCC_{healthy}| \leq 0.30$, $|PCC_{AS}| \geq 0.90$, $|PCC_{AS - healthy}| \geq 1.00$). (E) Visualization of the immune-related anti-function network of ABPA patients ($|PCC_{healthy}| \geq 0.70$, $|PCC_{ABPA}| \geq 0.70$, $|PCC_{ABPA - healthy}| \geq 1.00$). (F) Visualization of the immune-related anti-function network of asthma patients (($|PCC_{healthy}| \geq 0.70$, $|PCC_{AS}| \geq 0.70$, $|PCC_{AS - healthy}| \geq 1.00$)).

**Table 3 Hub nodes of the common immune-related functional network in ABPA and asthma.**

| Transcript ID | Gene name | Type | Differential expression of regulation in asthma | Differential expression of regulation in ABPA | Frequency of counting as top 20 hub nodes among 12 algorithms | P value |
|---|---|---|---|---|---|---|
| ENST00000327423 | PRR14L-201 | mRNA | UP | UP | 8 | 0 |
| MSTRG.89849.1 | MSTRG.89849.1 | Other | DOWN | DOWN | 8 | 0 |
| ENST00000606802 | AL139423.1-201 | lncRNA | DOWN | DOWN | 8 | 0 |
| ENST00000477475 | SRC-208 | Other | UP | DOWN | 8 | 0.98 |
| ENST00000399036 | CCRL2-202 | mRNA | UP | DOWN | 8 | 0.98 |
| MSTRG.38792.1 | MSTRG.38792.1 | Other | UP | DOWN | 8 | 0 |
| ENST00000369443 | GDAP2-202 | mRNA | UP | UP | 8 | 0 |
| MSTRG.68250.1[3] | MSTRG.68250.1 | Other | UP | DOWN | 8 | 0 |
| MSTRG.33289.1 | MSTRG.33289.1 | Other | DOWN | DOWN | 8 | 0 |
| ENST00000520643 | SPAG1-206 | mRNA | DOWN | DOWN | 7 | 0 |
| MSTRG.520.10 | MSTRG.520.10 | Other | DOWN | DOWN | 7 | 0 |
| MSTRG.69149.1[3] | MSTRG.69149.1 | Other | UP | DOWN | 6 | 0 |
| ENST00000595806 | HNRNPUL1-210 | lncRNA | DOWN | DOWN | 6 | 0 |
| ENST00000614509 | AC106028.4-201 | lncRNA | DOWN | DOWN | 6 | 0 |
| ENST00000619636 | SLC35B2-207 | mRNA | UP | DOWN | 6 | 0 |
| ENST00000531995 | PUF60-218 | lncRNA | UP | DOWN | 5 | 0 |
| ENST00000514385 | FXYD2-204 | Other | UP | DOWN | 5 | 0 |
| MSTRG.61564.1 | MSTRG.61564.1 | Other | UP | DOWN | 5 | 0 |
| ENST00000490736 | UEVLD-206 | Other | UP | DOWN | 4 | 0 |
| MSTRG.111947.1 | MSTRG.111947.1 | Other | UP | DOWN | 4 | 0 |
| ENST00000469356 | SREBF1-208 | lncRNA | DOWN | DOWN | 4 | 0 |
| MSTRG.96227.1 | MSTRG.96227.1 | Other | UP | DOWN | 3 | 0 |
| MSTRG.68251.1[3] | MSTRG.68251.1 | Other | UP | DOWN | 3 | 0 |

**Note:**
[3] The transcript exists in the cluster 3 of cluster analysis in Fig. 5E.

network can be categorized into four main immune-related groups, including leukocyte involved immune process, cytokine related immune process, autophagy apoptosis regulation and response to exogenous invasion. Topological analysis identified many key mRNAs/lncRNAs implicated in the regulation of diverse immune-related processes in response to ABPA and asthma (the frequency of counting as top 20 hub nodes among 12 algorithms ≥ 3, Table 3, details see Supplementary Methods), including five lncRNAs, namely AL139423.1-201, AC106028.4-201, HNRNPUL1-210, PUF60-218 and SREBF1-208, and five known mRNAs (PRR14L-201, CCRL2-202, GDAP2-202, SPAG1-206, SLC35B2-207). Notably, many mRNAs have already been demonstrated to be associated to asthma or ABPA by previous studies. For instance, Inhibition of PARP14 (Top one hub node in Table 3, it was found to significantly up-regulated in both diseases) was reported to reduce allergic airway diseases, it thereby was proposed as potential therapeutic target for asthma (*Mehrotra et al., 2013*).

Moreover, all the RNA–RNA interactions involved the five key lncRNAs were observed in the loss-of-function networks in both diseases and exhibited down-regulated expression

in the peripheral blood of patients in both diseases (Table 3), suggesting the regulations of hub lncRNAs in these interacted target RNAs were disrupted after the occurrence of asthma or ABPA. Their targets were found to be involved in many immune-related processes and pathways (Figs. 6A–6E). Permutation test shows majority of hub nodes are high-confidence (for details see Supplementary Methods). These findings indicate that the expression of these five lncRNAs would be repressed during the occurrences of ABPA or asthma, many immune-related processes mediated by them might be disrupted, simultaneously.

### Validation of hub nodes using GEO datasets

Four hub nodes including two mRNAs (ENST00000327423, ENST00000369443) and two lncRNAs (ENST00000587128, ENST00000595806) from the two networks were selected for validation. The expression profiles of four GEO datasets (GSE35571, GSE473, GSE31773 and GSE2125) were used. The results indicated that the expression level of the selected hub nodes are consistent with the result of GEO datasets (Figs. 6F–6I). Also, many interactions involved in these hub nodes could be found in these GEO datasets (Table S9), implying our predictions are reliable and convincing.

## DISCUSSIONS

ABPA has not received the importance that it deserves despites it occurs with a world-wide distribution in a great number of patients with asthma or cystic fibrosis (CF). Although more than half of century has passed since ABPA was first described, its exact pathogenesis as well as potential relationship with asthma is still unknown. The clinical data of the collected subjects in the present study do not show significant difference between ABPA and asthmatic patients, except that ABPA have higher IgE level (Table 1). We thereby tried to find some clues from the transcriptome. Initially, we looked into whether there is mutational difference between ABPA, asthma and health controls. The results showed that many genes (including two lncRNAs: LINC00398, LINC00892) exhibited distinct mutations across those samples (Fig. 2A). Regarding SSRs, no difference in biological significance was found cross those samples, despite some of them showed significant $P$ value in statistic test after filtering. These findings indeed are consistent to the mainstream view on the pathophysiological mechanisms of ABPA and asthma, i.e., unlike genetic diseases that mutations largely contribute to the pathogenesis of these diseases, the development of ABPA or asthma should involve the complex cross-talking between specific heredities and environmental factors. Mutations particularly in transcriptome might have only a marginal effect on the occurrence of both diseases. This finding actually is consistent with other previous studies, which indicated that genetic risk factors have overall small effects for the adult-onset asthma, implying a greater role for non-genetic risk in adult-onset asthma (Pividori et al., 2019).

Next, we tried to find some clues from the comparison of gene expression among those groups, particularly in lncRNAs. In addition to traditional differential expression

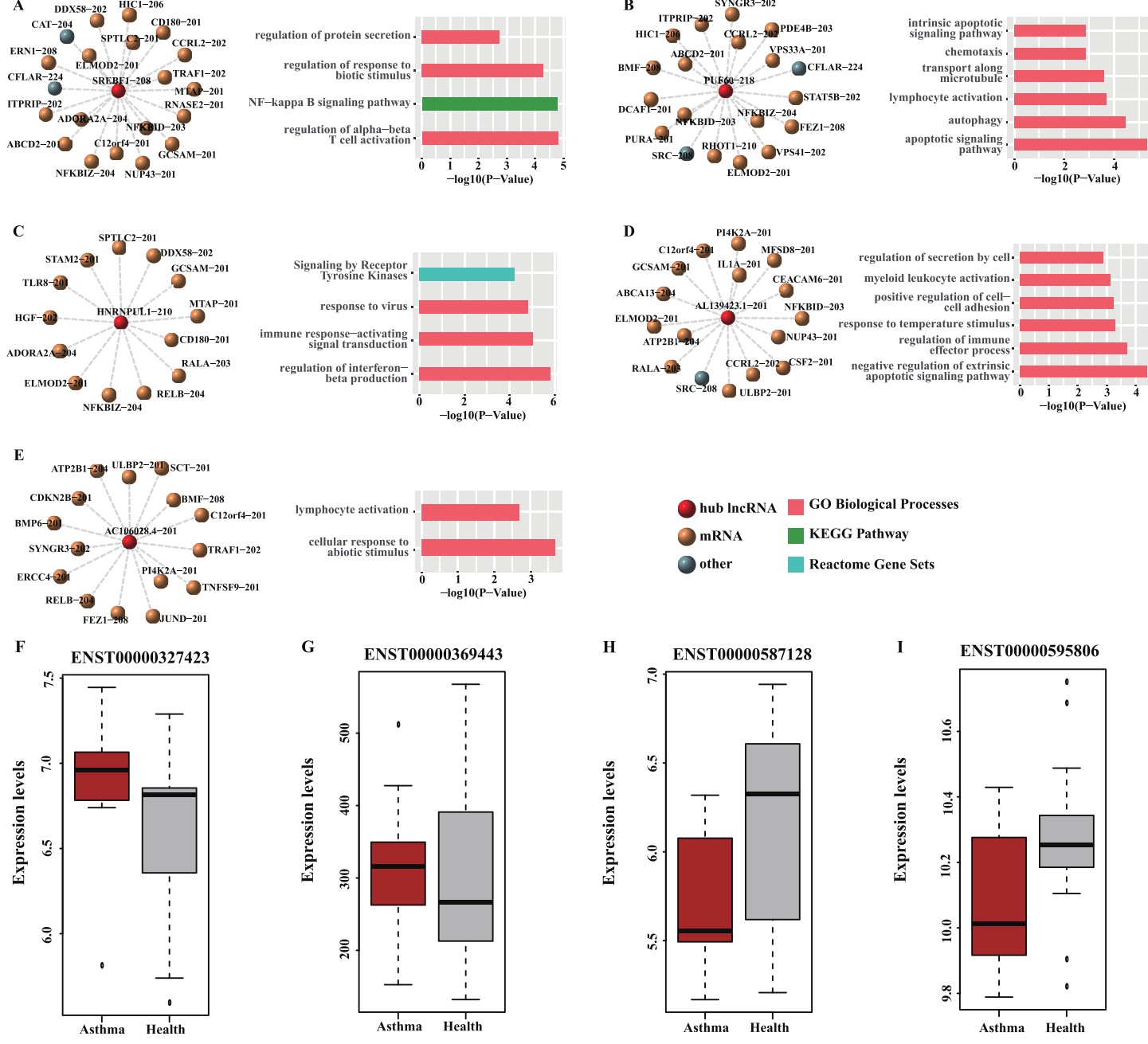

**Figure 6 Functional analysis and GEO validation of key lncRNAs (hub nodes) derived from the loss-of-function networks of two diseases.** (A) hub node: SREBF1-208. (B) hub node: PUF60-218. (C) hub node: HNRNPUL1-210. (D) hub node: AL139423.1-201. (E) hub node: AC106028.4-201. (Left) sub-networks display all the target mRNAs for the key lncRNAs. (Right) bar plots show Gene ontology (GO) functional enrichment analysis based on the corresponding target mRNAs. (F–I). Validation of the expression level of four selected hub nodes (ENST00000327423, ENST00000369443, ENST00000587128 and ENST00000595806) between asthmatic groups and health controls. (F) Validation based on microarray data of GSE35571. (G) Validation based on microarray data of GSE473. (H) Validation based on microarray data of GSE31773. (I) Validation based on microarray data of GSE2125.

analysis, we proposed a differential network analysis to detect the dynamic changes of transcriptome across those different groups. Differential network analysis based on three different filtering criteria figured out several key lncRNAs and mRNAs. The reason that we
applied different filtering criteria for three different groups is that three networks exhibit extremely distinct threshold value of pearson correlation coefficient, and different criteria would yield a similar number of RNA–RNA interactions in different groups. Notably, we applied 12 different algorithms of topological network analysis to make sure our predictions more reliable and convincing. Our analysis indicated that the interactions involved in 82 key transcripts (hub nodes in network) were dramatically disrupted by the occurrence of ABPA, and some interactions were repressed by ABPA and some were gained in response to ABPA (Table S7). For instance, the gene RGS3 (Top 2nd hub node in Table S7) encodes the regulator of G-protein signaling three, and it can activate MAP kinases (MAPK), which was found to be critically involved in modulation of asthmatic inflammation. It also was proposed for a potential therapeutic target for asthma (*Huang et al., 2019b*). The gene GPR171 (Top 3rd hub node in Table S7) encodes probable G-protein coupled receptor, which was reported to be implicated in asthma endophenotypes and negatively regulate myeloid cell differentiation (*Thompson et al., 2006*). Similarly, we identified 93 key mRNAs/lncRNAs (Table S8) in asthma network. Few hub nodes share in ABPA and asthma networks, suggesting that these distinct hub nodes might be good candidates for further investigating the difference between ABPA and asthma. Some mRNAs/lncRNAs were found as new risk factors associated with ABPA or asthma, i.e., AL139423.1-201, GDAP2, which might be novel biomarkers and targets for diagnosis and therapy in both diseases.

By integrating two networks, we could find several common key mRNAs and lncRNAs. Permutation test shows the reliability of this analysis. Some of them have already been demonstrated to be correlated to asthma or ABPA. For example, PARP14 was found to play a role in transcription of interleukin-4 (IL4)-responsive genes, which controls cell survival, metabolism and proliferation (*Cho et al., 2009*). Inhibition of PARP14 was found to reduce allergic airway diseases (38). CCRL2 (Top 4th hub node in Table 3) encodes a chemokine receptor like protein. Chemokines and their receptors have reported to mediate signal transduction, which are critical for the recruitment of effector immune cells to the site of inflammation (*D'Ambrosio et al., 2001*). More importantly, many known and novel lncRNAs were identified to be related to asthma and ABPA for the first time, i.e., AL139423.1-201, HNRNPUL1-210, AC106028.4-201, etc.

## CONCLUSIONS

Our analysis discloses many lncRNAs associated to immune-related mRNAs, which suggests these lncRNAs might participate regulation of immune-related processes in response to the occurrence of both diseases. Further investigation on the interactions among them might provide some clues to access underlying mechanisms of pathogenesis for both diseases. Certainly, more ingenious experimental design and validation is required to conclude the concrete roles of these lncRNAs in both diseases. In short, our analysis describes a rough transcriptome landscape of ABPA and asthma, and benefits understanding the pathogenesis of both diseases.

### Funding

This work was funded by the University of Macau (Grant Numbers: FHS-CRDA-029-002-2017, EF005/FHS-ZXH/2018/GSTIC, MYRG2018-00071-FHS and SRG2016-00083-FHS) by The Science and Technology Development Fund, Macau SAR (File No. 0004/2019/AFJ), the Guangdong Science and Technology Fund (Project No.: 2020B1111300001), and the Guangzhou Institute of Respiratory Health Open Project (Funds provided by China Evergrande Group) (Project No.: 2020GIRHHMS04). The funders had no role in study design, data collection and analysis, decision to publish, or preparation of the manuscript.

### Grant Disclosures

The following grant information was disclosed by the authors:
University of Macau: FHS-CRDA-029-002-2017, EF005/FHS-ZXH/2018/GSTIC, MYRG2018-00071-FHS and SRG2016-00083-FHS.
Macau SAR: 0004/2019/AFJ.
Guangdong Science and Technology Fund: 2020B1111300001.
Guangzhou Institute of Respiratory Health Open Project: 2020GIRHHMS04.

### Competing Interests

The authors declare that they have no competing interests.

### Author Contributions

- Chen Huang analyzed the data, prepared figures and/or tables, authored or reviewed drafts of the paper, and approved the final draft.
- Dongliang Leng analyzed the data, prepared figures and/or tables, authored or reviewed drafts of the paper, and approved the final draft.
- Peiyan Zheng performed the experiments, authored or reviewed drafts of the paper, and approved the final draft.
- Min Deng analyzed the data, prepared figures and/or tables, and approved the final draft.
- Lu Li performed the experiments, authored or reviewed drafts of the paper, and approved the final draft.
- Ge Wu performed the experiments, prepared figures and/or tables, and approved the final draft.
- Baoqing Sun conceived and designed the experiments, authored or reviewed drafts of the paper, and approved the final draft.
- Xiaohua Douglas Zhang conceived and designed the experiments, authored or reviewed drafts of the paper, and approved the final draft.

### Field Study Permissions

The following information was supplied relating to field study approvals (i.e., approving body and any reference numbers):

This study was approved by the Medical Ethics Committee of First Affiliated Hospital of Guangzhou Medical University (ethics approval no. gyfyy-2016-73). All experiments were

performed in accordance with relevant guidelines and regulations of the Ethics Committee of First Affiliated Hospital of Guangzhou Medical University.

## Data Availability

All the raw RNA-seq data used in the present study are available at NCBI SRA: PRJNA582337. All the clinical raw data for the statistical analysis in Table 1 are available in the Supplemental File.

## Supplemental Information

Supplemental information for this article can be found online at http://dx.doi.org/10.7717/peerj.11453#supplemental-information.

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
