# Peer review of "Comprehensive transcriptome analysis of peripheral blood unravels key lncRNAs implicated in ABPA and asthma"

_PeerJ, doi:10.7717/peerj.11453_

## Round 0.1 · original submission · Major Revisions

In addition to the constructive and objective comments from the three reviewers, I have the following concerns regarding your manuscript:

1. The sample size of four GEO validation sets should also be provided. And the authors should clarify whether lncRNA or mRNA was validated in these datasets.

2. What are the criteria for the selection of four hub nodes (Line 257)? Since five lncRNAs and five mRNAs have been confirmed or found, why only verify these four hub nodes? Moreover, the main topic of the paper should be lncRNA, so why only two lncRNAs were validated?

3. The names of the lncRNA or mRNA are not uniformed throughout the whole manuscript. Some places use Transcript IDs, and other places use gene names, which is confusing. Please use the same name consistently.

4. GO items represented by different colors in Figures 3B and 3C. What is the significance of the use of these different colors?

5. The discussion section could be written better. It seems to be just a rearrangement of the Results section. Normally in this section, important findings are needed to be further discussed in combination with the literature.

6. What are those 12 algorithms (Line 239 and 282)? It should be better to indicate the specific algorithm names in the paper.

When revising your manuscript, please consider all issues mentioned in the comments from the two reviewers carefully and provide suitable responses for any comments. Please note that your revised submission may need to be re-reviewed.

PeerJ values your contribution and I look forward to receiving your revised manuscript.

Reviewer 1 ·

Basic reporting

Huang et al. performed a comprehensive transcriptome analysis of peripheral blood in order to unravel key lncRNAs implicated in ABPA and asthma.The authors identified high-confidence lncRNAs from the RNA-seq data from three types of human peripheral blood.Then they integrated a few transcriptome analysis methods to identify crucial lncRNAs. Differential expression analysis and differential network analysis were performed to conform and filter the important candidate lncRNA. The study is well organized.However, there are some specifice comments:

There are some obvious typos, even in the Abstract.For example, “Our analysis indicated that these lncRNAs exhibits in the loss-of-function
networks”and “a high-confifidence dataset of lncRNAs were identifified using a stringent fifiltering pipeline”. So I suggest the authors to go over the text carefully.

Experimental design

Before the analysis, the authors should provide some indexes to show the quality of sequencing data, such as mapping rate.
The threshold values of the correlation coefficients are totally different in three types of networks. The analysis process is somewhat subjective. Suggest the authors to show some evidence to support your selection.

Validity of the findings

According to my understanding, the analysis of this study is based on the RNA-seq data.It is a little confused that some variants (~26,。2% )via GATK analysis pipeline located in intergenic region.

Reviewer 2 ·

Basic reporting

This paper was overall well written, with sufficient data analysis and proper statistics.

Experimental design

The experimental design is OK, with 2 disease groups, and 1 control group.

Validity of the findings

The methods part includes clear information. Literature reviews also validate some of their findings. However, more details should be added to their validation parts.

Additional comments

Review of Manuscript 56161, " Comprehensive transcriptome analysis of peripheral blood unravels key lncRNAs implicated in ABPA and asthma"

This paper analyzed the RNAseq data of PBMC from a total of 27 unique individuals, including 7 asthma patients, 12 ABPA patients and 8 healthy individuals considered as control group. The raw data was mapped to human genome to get lncRNAs, including unknown lncRNAs through a series of careful designed procedures. In addition, the sequence variances were also identified from the RNAseq sequences using GATK tools. Then, both the genetic variances and expressions among these 3 groups were systematically analyzed. Immune related modules were enriched from the DETs among these 3 groups. After that, the authors built a network between the DETs and other transcripts, network alterations among these 3 groups were identified to find sub-networks for each disease, namely loss-of-function network, gain-of-function network, and anti-function network. Finally, network hubs were analyzed and 4 of these nodes were chosen for validation.

Main comments:
1. Line 151, please add a few details of how GlueGO work. Is the function enrichment based on differentially expressed genes based on a certain p value? Or it was ranking based/GSEA-style enrichment?
2. Line 154-155, please also explain why the correlations between DETs were not calculated and compared between groups.
3. Line 188, how did the authors perform the “across samples by Fisher Exact probability test”? From the supplementary file, they were using Fisher Exact test. In this case, it should be “across groups” rather than “samples”.
4. Line 207, how did the authors perform “cluster analysis”. Cluster analysis based on what? They should write it clearly to make the readings more smoothly.
5. In the validation part, the authors should explain why they chose those 4 transcripts for validation purposes. Also, they validated the results from 4 different GEO datasets, please explain why those 4 datasets were chosen. Also, did they validate the 4 transcripts in all 4 datasets? Or they validate them as one dataset for one transcript? Why not using one convincing dataset to validate all transcripts? They may provide a Venn diagram of the DETs of the other dataset and their hub genes.

Minor points:
1. Line 119, should be “Figure 1”, rather than “Figure S1”. Please check.
2. There are some other typos, such as in Line24, “It complicates and aggravate” should be “It complicates and aggravates”… etc. please read carefully and check.
3. In figure 5, please add a title to each network.
4. In table1, sex information should also be included to exclude the co-variance effects.

Reviewer 3 ·

Basic reporting

The introduction are very clear and written well.

Experimental design

Number of samples are insufficient for this analysis.As per data, less than 1%of asthma patient were affected by ABPA.So to predict the analysis ,the sample number should be high.
Also , Author didn't mention the type or stage of the asthmatic patient.The experiment is not well designed.

Validity of the findings

The findings are valid.But to prove the conclusion the sample number of asthama,ABPA and control should be high.Also the stages of the disease has to be noted clearly.

Additional comments

The author work is done very informatively and clearly.I really appreciate the author research.I suggest author may also include cystic fibrosis patients.

---

## Round 0.2 · Minor Revisions

I am pleased to inform you that your submission entitled “Comprehensive transcriptome analysis of peripheral blood unravels key lncRNAs implicated in ABPA and asthma” has been provisionally accepted for publication in PeerJ.

However, before your paper can be forwarded to our Production Department, you are requested to make the corrections indicated below and also according to the suggestions from Reviewer 2.

(1) The sub-figures in Figure 4 (Figure 4A-Figure 4F) are not correctly cited in the texts between line 220 and line 225.
(2) The supplementary files do not include Figure S3 which is cited in the text in line 288.

We look forward to receiving your final version of your manuscript.

Reviewer 1 ·

Basic reporting

no comment

Experimental design

no comment

Validity of the findings

no comment

Reviewer 2 ·

Basic reporting

In the revised manuscript, the authors addressed all my concerns.

Experimental design

no comment

Validity of the findings

no comment

Additional comments

In the revised manuscript, the authors addressed all my concerns.
minor concerns:
1. In line 228, still "across samples" though the author claimed that was changed in the rebuttal letter.
2. In table S2, could the authors also add one column that showed some basic information of the GEO datasets? Should include the basic experimental design, types of sequencing(RNAseq vs microarray), the tissue where RNA was extracted, etc.

---

## Round 0.3 · accepted · Accept

Thank you for the high quality article you provided.